# Engineering the temporal dynamics of all-optical switching with fast and slow materials

Soham Saha[1,2], Benjamin T. Diroll [2], Mustafa Goksu Ozlu[1], Sarah N. Chowdhury[1], Samuel Peana[1], Zhaxylyk Kudyshev[1], Richard D. Schaller [2], Zubin Jacob [1,3], Vladimir M. Shalaev [1,3], Alexander V. Kildishev[1,3] & Alexandra Boltasseva [1,3] ✉

All-optical switches control the amplitude, phase, and polarization of light using optical control pulses. They can operate at ultrafast timescales – essential for technology-driven applications like optical computing, and fundamental studies like time-reflection. Conventional all-optical switches have a fixed switching time, but this work demonstrates that the response-time can be controlled by selectively controlling the light-matter-interaction in so-called fast and slow materials. The bi-material switch has a nanosecond response when the probe interacts strongly with titanium nitride near its epsilon-near-zero (ENZ) wavelength. The response-time speeds up over two orders of magnitude with increasing probe-wavelength, as light's interaction with the faster Aluminum-doped zinc oxide (AZO) increases, eventually reaching the picosecond-scale near AZO's ENZ-regime. This scheme provides several additional degrees of freedom for switching time control, such as probe-polarization and incident angle, and the pump-wavelength. This approach could lead to new functionalities within key applications in multi-band transmission, optical computing, and nonlinear optics.

Switchable optical devices enable real-time control over the polarization, amplitude, or phase of light[1–7]. In all-optical switching, a light pulse interacts with the device's constituent material, changing its optical response. For example, an intraband pump can energize free electrons in a transparent conducting oxide (TCO), increasing their effective mass, making the materials less absorptive[8–12] and thus increasing overall transmission. Conversely, an interband pump can generate photocarriers that increase the metallicity and absorption of TCOs[13,14]. This mechanism is often suggested for dynamic photonic devices capable of modulating the amplitude or phase of light passing through them. Importantly, all-optical switches can operate without the resistive-capacitive delays of electronic circuits. Instead, the speed of optically-induced permittivity modulation is limited by the relaxation mechanisms of the switching material, which can range from a few femtoseconds[15,16] to several nanoseconds[17,18]. The fast permittivity change enables several interesting phenomena and applications that are not achievable by other means of switching. Large and rapid modulation of the dielectric permittivity in TCOs has led to non-reciprocal optical devices[19], photon acceleration[20], and ultrafast optical switching[21,22], to name a few examples.

Recently, field-enhanced slow-light effects in epsilon-near-zero materials, including TCOs, have further enhanced these nonlinear optical phenomena by several orders of magnitude[23,24]. The wavelength range where the real part of the material dielectric permittivity changes its sign is known as the Epsilon-Near-Zero regime. Strong light-matter interaction due to the large field enhancement and the slow group velocity of light near the ENZ point enables a plethora of interesting optical phenomena without the need for complex composite structures. Examples of such phenomena include dramatic reflectance and transmittance modulation[25], remarkably strong

[1]School of Electrical and Computer Engineering, Birck Nanotechnology Center, Purdue University, West Lafayette, IN, USA. [2]Argonne National Laboratory, Lemont, IL 60439, USA. [3]Purdue Quantum Science and Engineering Institute, Purdue University, West Lafayette, IN, USA. ✉e-mail: aeb@purdue.edu

nonlinearity enhancement[26–28], time refraction[29], broadband[30] and narrowband[31] absorption, optical time reversal[32], and high-harmonic generation[33], in addition to on-chip modulators[34,35]. When an ENZ medium is fabricated on a reflective metal, free-space light can couple into radiative bulk plasmon-polaritons, called Ferrell-Berreman (FB) modes[36,37]. Strong light-matter interaction in such multilayer cavities has been employed to demonstrate broadband absorbers[30] and polarization switches[9].

Any optical switching device relies on two major functionalities: the magnitude of the dielectric permittivity change, which governs the modulation depth, and the overall material response time that governs the switching speed. There have been numerous studies on modulating the permittivity of materials in various ways, including electrical[35,38–41], thermal[42–45], and optical methods[9–11,14,46–54]. These studies have focused on the scaling of modulation with power[26,55], enhancement with engineered structures[8,14], and modulation limits[12,56,57].

Thus far, few studies have focused on engineering the switching speed of tunable and tailorable devices[58], which is generally fixed and/ or defined by the response times of the material constituents. The challenge in controlling the speed of all-optical switches is that the relaxation time of the switching material is an intrinsic property. As a rule, this property can be changed or adjusted at the film growth/ fabrication step but not dynamically during device operation. Thus, the overall switching response is assumed to be fixed after the device is fabricated. In one such study, the relaxation time of cadmium oxide-based switches was decreased by increasing the doping concentration[11]. Epsilon-near-zero switches utilizing aluminum-doped zinc oxide have a picosecond response[13] when controlled by an interband pump, but a femtosecond response[15] when controlled by an intraband pump. All-optical switches with higher quality factors generally have a longer response time than those with a lower quality factor[59]. The carrier dynamics of gold nanorods can be somewhat modified by pumping them at varying wavelengths or angles, varying the switching time to a limited degree[60]. Overall, the range of switching times obtained in these prior studies are within the same timescale order, determined by the carrier dynamics in the single switching material.

Interestingly, several works studying the carrier dynamics of different materials report wavelength-dependent response times. For example, in pump-probe studies of gold, longer wavelength probes exhibit faster relaxation times as they interact with states that relax faster than those probed by shorter wavelengths[61]. In black phosphorus, for the same pump wavelength, the decay time of sub-bandgap probes decreases at shorter wavelengths[62]. In titanium nitride, a sub-picosecond electron-phonon response is only observed when the probe wavelength is close to the epsilon-near-zero (ENZ) wavelength[18]. These experiments highlight that the device operational wavelength has an important role in the observed dynamics of a structure. Extending this idea, we demonstrate a device-level approach to controlling all-optical switching speed in a dual-active-material structure employing the Epsilon Near Zero properties of the materials.

Here, we show the utilization of two technologically relevant, robust materials with extensively studied relaxation dynamics: plasmonic titanium nitride (TiN) and aluminum-doped zinc oxide (AZO), in a single multi-resonant device. Individual transient characterization of TiN demonstrates an overall relaxation time spanning nanoseconds[17], while that of AZO shows a much faster relaxation time spanning picoseconds[13,15]. Our approach for engineering the temporal dynamics involves combining these materials, resulting in a double-resonant device that supports a radiative ENZ mode in the TiN layer and a Ferrell Berreman mode in the AZO layer[36,63]. Upon using a 325-nm-wavelength optical pump, electrons in both TiN and AZO are simultaneously excited, leading to transient reflectance modulation due to changes in permittivity. Notably, the reflectance modulation is most pronounced

near the epsilon-near-zero regions of the corresponding materials. The device switching time is nanosecond-speed in proximity to the ENZ wavelength of TiN, following its carrier dynamics, and picoseconds, close to the ENZ of AZO, consistent with AZO dynamics. This highlights the significant influence of light-matter interaction with distinct materials in an optically modulated device. Moreover, we establish a model for the temporal dynamics of devices employing multiple functional materials. We further demonstrate that the device switching time can be controlled utilizing a multitude of parameters, including the angle of incidence and the polarization of probe beam, and the pump-wavelength configurations. Ultimately, we discuss the experimental findings and their implications for designing more complex all-optical switches.

## Results

We developed a TiN-AZO FB resonator by depositing a 130-nm-thick layer of TiN on a silicon substrate, followed by the growth of a 250-nm-thick AZO layer (Fig.1a). Supplementary Information Sections 1–3 elaborate on the growth parameters and the optical characterization of the fabricated films. For our pump-probe experiments, we use a normal-incidence 325-nm-wavelength pump beam, which is absorbed in the AZO and the TiN layers. This effect is evident from the absorbance plots obtained from finite-element frequency domain (FEFD) simulations (COMSOL Multiphysics) (Fig. 1b, left). The probe light is shone at a 50° angle of incidence. At visible wavelengths, the probe mostly interacts with the TiN, whereas at near-infrared wavelengths, it interacts strongly with the AZO (Fig. 1b, right). The structure shows two reflectance dips for p-polarized light near the ENZ points of TiN and AZO, evident in Fig. 1c.

In the fabricated TiN-AZO resonator, the 325-nm pump excites electrons in TiN and AZO, red shifting the resonance in the visible[18], and blue-shifting the resonance in the near-infrared (NIR) range[13] (Fig. 1c). Reflectance modulation occurs at nanosecond timescales for visible wavelengths and speeds up as the wavelength increases, until it reaches picosecond timescales at NIR wavelengths (Fig. 1d).

This transition happens because, in the visible regime, the probe light is mostly localized in and interacts strongly with TiN, thus the slow carrier dynamics of TiN dominate the dynamics of the switch. Similarly, near the ENZ point of AZO at near-infrared wavelengths, AZO dynamics drives the switching speed. The switching speed between the two material resonances lies somewhere between the TiN and AZO relaxation times. The speed increases as the probe wavelength gets closer to the AZO resonance. Thus, by exploiting the nonlinearities of the two materials, our device can operate at speeds ranging from the GHz to the THz regime, differing by two orders of magnitude with the same optical pump.

The next sections describe the design and characterization of the hybrid device. We start with the steady-state reflectance characterization to determine the nature of the observed resonant dips.

### Epsilon-near-zero enhanced all-optical switch design and fabrication

A typical Ferrell-Berreman resonator, which comprises an epsilon-near-zero medium with a subwavelength thickness on a reflective backplane[36,63], is engineered to directly couple free-space p-polarized light into the medium. Figure 2a shows the device schematic. We perform reflectance measurements on our TiN-AZO double-layer at the 50° angle of incidence. The device shows two dips near the TiN and AZO ENZ points in the reflectance spectrum of p-polarized light (Fig. 2b). Around 1360 nm p-polarized light near the AZO ENZ couples into a radiative Berreman mode, resulting in a reflectance dip. TiN is metallic near the ENZ of AZO, serving as the back reflector required for the Berreman mode. On the other hand, near the ENZ of TiN, AZO is a dielectric allowing light to pass into the TiN. The high refractive index of silicon allows it to act as a reflective backreflector, allowing light to

couple into the radiative ENZ mode[64] in the TiN, resulting in the reflectance dip ~485 nm. Thus, our device supports a radiative ENZ mode in the visible and a Ferrell-Berreman mode in the infrared. The *s*-polarized light spectra have multiple Fabry-Perot resonance dips throughout the wavelength range measured. Spectroscopic ellipsometry followed by fitting with a Drude-Lorentz model shows that TiN has an epsilon-near-zero point at 485 nm, and AZO at 1360 nm (Fig. 2c, d). Supplementary Table S1 has the DL parameters used in this study.

The planar structure of this dual-resonant device also ensures that additional nanostructuring-induced recombination channels[14,65] do not affect its switching response.

## Pump-probe spectroscopy of individual films

To understand the effect of each film's temporal response on the overall behavior of the device, we pre-characterized the TiN and AZO films, individually grown on silicon, close to their respective ENZ points. Figure 3a shows the experimental setup used to measure the response of TiN on silicon. An 800 nm, 2 kHz amplified Ti: sapphire laser is split into two branches a pump branch and a probe branch. The pump branch is directed into an OPA (Topas) and converted to 325 nm. The probe branch is delayed using an electronically controlled delay stage and then focused into a Beta Barium Borate (BBO) crystal to generate supercontinuum white light. The pump and probe beams are spatially overlapped on the sample. The pump-fluence was fixed at 1.5 mJ/cm²/pulse.

The temporal response of charge carriers in TiN has been studied in detail by George et al.[17] and Diroll et al.[18] Upon excitation by an optical pump, the reflectance spectrum of the TiN redshifts. The redshift results in broadband reflectance modulation, with a positive reflectance change to the left of the minimum and a negative change to the right. In our pump-probe measurements, we see a similar TiN spectral response (Fig. 3b). Supplementary Information S4 has more details on the dynamic characterization of TiN films, Supplementary Fig. S1 has the experimentally obtained graphs and Table S2 contains the fitting parameters. The response can be accurately fitted with a two-time-constant model (Eq. S2), with the dynamics attributed to lattice cooling[18]. The overall response time of our TiN film is more than 1 nanosecond (Fig. 3c), which determines the relatively slow switching speeds of TiN mirrors.

Next, we studied the temporal response of AZO films under the same pump. We deposited a 300-nm-thick AZO film on silicon for these studies (see Supplementary Information for the growth and modeling details). We performed the pump-probe spectroscopy on the AZO using the setup shown in Fig. 3a, with a sapphire replacing the BBO crystal to produce the supercontinuum NIR probe (800–1600 nm). The pump was formed using the same method described in the previous section.

The interband pump generates free carriers in AZO, increasing the absorption and decreasing the refractive index by modifying a Drude dispersion term[13]. The reflectance spectrum blueshifts, decreasing the reflectance to the left of the minima and increasing to the right (Fig. 3d). As electrons recombine, the sample returns to its original refractive index. Supplementary Information S5 has more details on the dynamic characterization of AZO films. Supplementary Fig. S2 has the experimentally obtained graphs and Table S3 contains the fitting

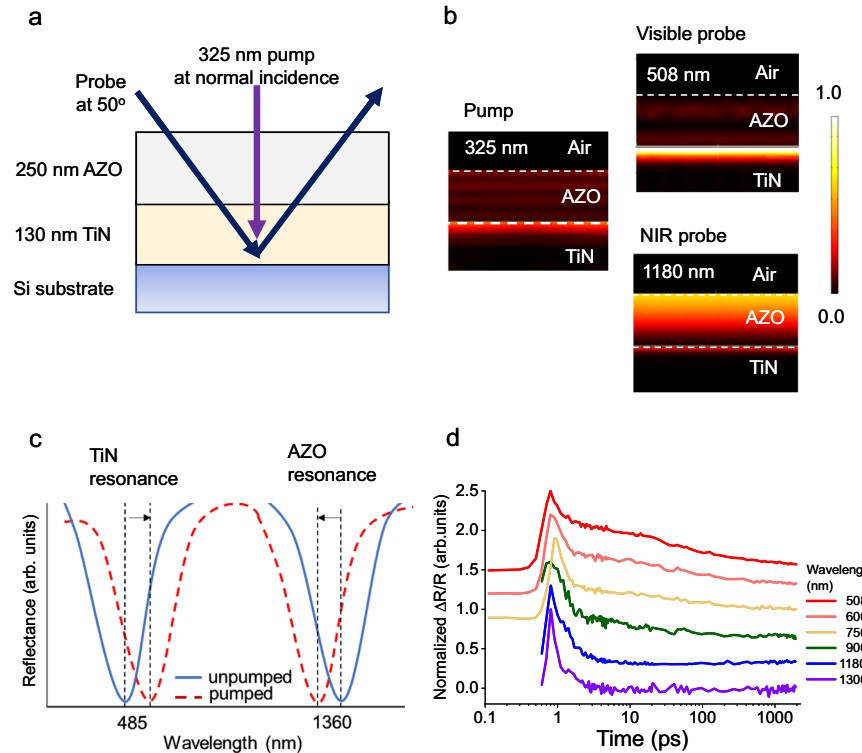

**Fig. 1 | Basic principle for adjustable switching dynamics. a** The double-resonant two-layer device comprises a 130-nm-thick TiN layer grown on silicon, with a 250-nm-thick AZO layer deposited on top. **b** Normalized power dissipation density of probe in the different layers simulated by COMSOL Multiphysics. At normal incidence, the 325-nm wavelength pump is strongly absorbed in the AZO and TiN by exciting electrons in both materials. The materials interact most strongly with light near their respective ENZ wavelengths. Thus, at visible wavelengths, most of the probe interacts with the TiN, whereas the NIR probes interact more with the AZO layer. **c** The pump causes the reflectance spectrum to redshift at visible wavelengths, whereas at near-infrared wavelengths, the pump blueshifts the reflectance spectrum. **d** The mechanism of fast and slow switching: TiN has a nanosecond response time, and AZO a picosecond response time. When excited by the same pump, the device has a slower observed response time in the visible probe wavelengths, where its behavior is dominated by the TiN response. At increasing wavelengths, its response speeds up as the relative light-matter interaction of the probe with the AZO increases.

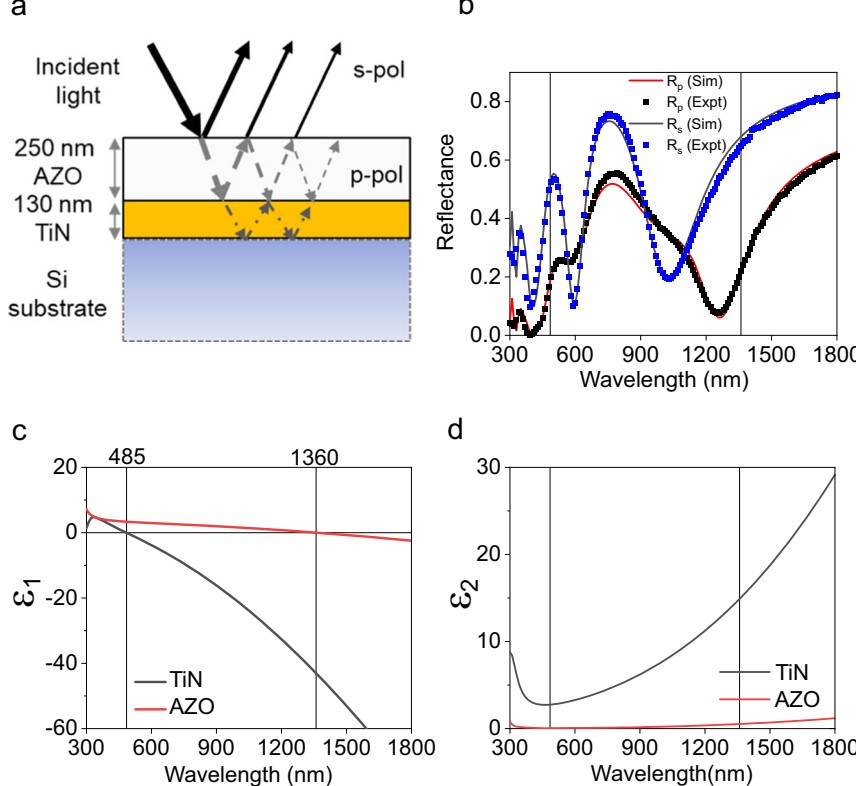

**Fig. 2 | Resonator schematic and material properties. a** Device schematic: The double-layer TiN-AZO resonator cross-section. **b** Reflectance spectra for the s- and p–polarized light with an incidence angle of 50°. Near the ENZ of the respective films, p-polarized light couples into either the radiative ENZ or FB mode.

S-polarized light can couple into Fabry-Perot modes, or is reflected. **c** The real part of the permittivity of the as-deposited TiN and AZO films; the vertical lines show the ENZ points of the films. **d** The imaginary permittivity of the films.

parameters. The transient reflectance can be fitted with a single sub-picosecond time constant (Eq. S3). This result points to a dominant, single recombination channel. In prior work, it was identified as the defect-assisted Shockley-Read-Hall mechanism[13]. The overall relaxation time shows little change if the pump fluence is varied. The reflectance modulation falls to zero in <10 ps (Fig. 3e), allowing for the very fast switching of AZO switches. We note that the substrate on which a conducting oxide is grown may affect the defect density or the switching response[66]. To account for this, we also grew a thick AZO film on fused silica and observed similar picosecond scale switching speeds (Supplementary Information S5).

**Pump-probe spectroscopy of the device**

After pre-characterizing the individual TiN and AZO films' dynamic response, we performed pump-probe measurements on the device in the reflection mode. The pump-probe setup is the same in Fig. 3a used for the individual film characterization. We used both visible and NIR light to probe the device. The pump fluence was kept at 1.5 mJ/cm², the same as in the previous section. The maximum reflectance modulation at visible and near-infrared wavelengths occurs proximal to the steady state-reflectance dips previously measured for individual TiN and AZO films. In particular, these dips occur near the ENZ points of the TiN and AZO films.

At visible wavelengths, there is an overall decrease in the reflectance above the minimum modulation wavelength and an increase below this wavelength (Fig. 4a) with a nanosecond response. The amplitude and the response dynamics are similar to the photoexcited TiN film in the previous section (Fig. 3b, c). The overall response time of the modulation is slow (ns range).

In the near-infrared regime, the dynamic spectral change is similar to that of AZO. The maximum change is negative below and positive above the modulation minimum wavelength (Fig. 4b). This corresponds to a transient blueshift in the reflectance dip with a picosecond response, similar to that of the AZO film (Fig. 3d, e). The relaxation time of the response is under 10 picoseconds.

Finite Element Method (FEM) simulations (COMSOL Multiphysics) with our material's measured optical properties show that at 325 nm (under normal incidence), 48% of the pump light is absorbed in the TiN layer and 39% in the AZO layer. Thus, upon excitation by the pump, electrons in both AZO and TiN are excited, modulating the permittivity of both materials. Carriers in AZO relax at ultrafast timescales while the permittivity of TiN takes longer to settle to steady-state due to its slower lattice cooling relaxation process.

FEM simulations (COMSOL) using the extracted optical properties of TiN and AZO show that 80% of p-polarized light is absorbed in the TiN near its ENZ point in the visible. Similarly, 90% of p-polarized is absorbed by the AZO near its ENZ point in the NIR (Fig. 4c). Thus, the visible probes interact strongly with the TiN layer near its ENZ, resulting in slower, nanosecond switching times. On the other hand, near its ENZ the near-infrared probe interacts strongly with the AZO film, resulting in ultrafast, picosecond scale switching time.

To further illustrate the phenomenon, in Fig. 4d we overlap the temporal response of the device and that of the individual thick films from the previous section at different wavelengths. At 508 nm, the modulation dynamics of the bilayer device (solid red line) closely follows that of the TiN film on silicon (red dots), as the probe strongly interacts with the titanium nitride layer (Fig. 4c). On the other hand, at 1180 nm, the probe strongly interacts with the AZO layer, as shown by

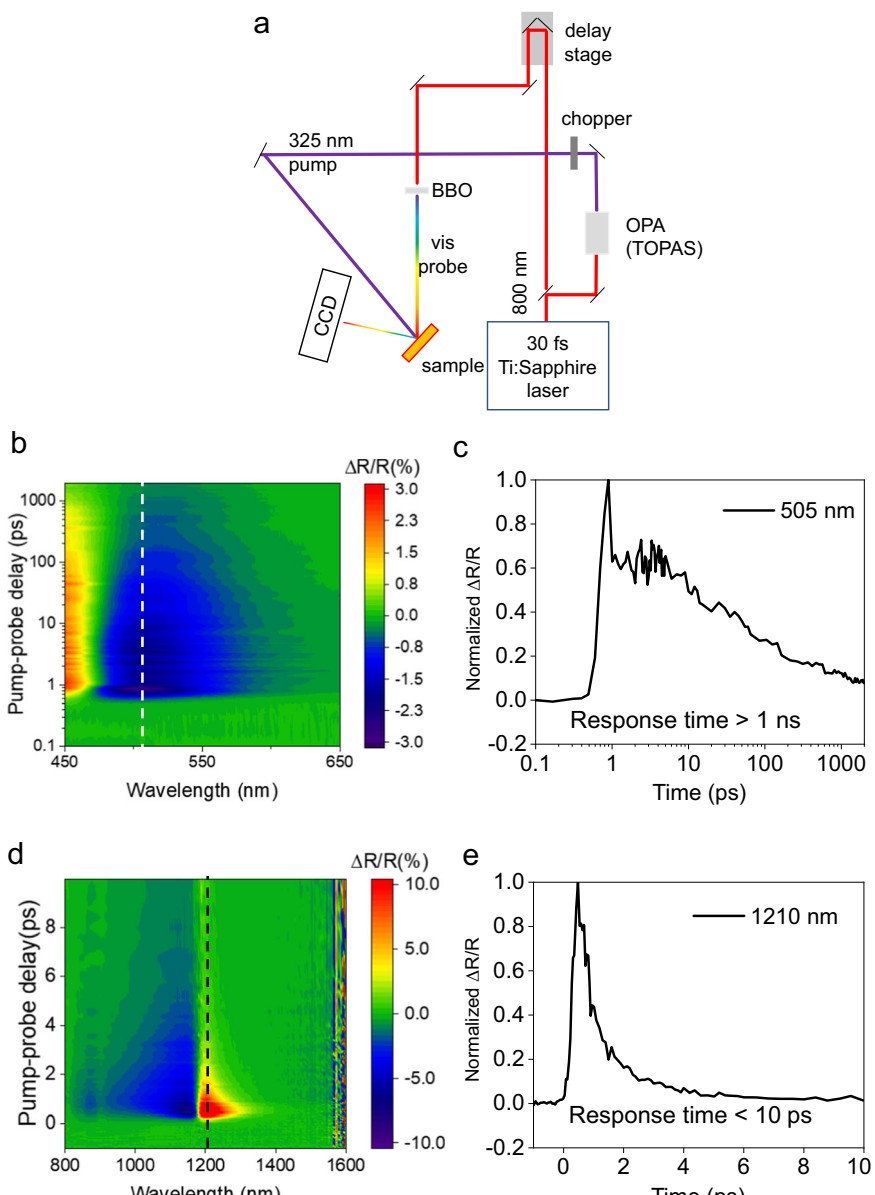

**Fig. 3 | Experimental setup, and pump-probe results of individual materials.**
**a** The pump-probe spectroscopy setup in reflectance mode with a 325 nm pump and visible probe. **b** Transient reflectance modulation versus time for TiN on Si probed at visible wavelengths. The response along the vertical dashed line near the ENZ wavelength of 505 nanometers is plotted in **c**. **c** The modulation has a zero-to-zero response time of several nanoseconds at 505 nm. **d** Transient reflectance modulation versus time for 500 nm AZO on silicon probed at near-infrared wavelengths. The response along the vertical dashed line at the near ENZ wavelength of 1210 nm is plotted in **e**. **e** The modulation has a zero-to-zero response time of <10 ps at 1210 nm.

the strong absorbance of *p*-polarized light in the AZO. As a result, the speed of the modulation at 1180 nm (solid blue line) mirrors that of AZO on silicon (blue dots).

**Effective temporal response of the device**

At wavelength regimes where the probe interacts comparably with TiN and AZO, the response has attributes of both materials. Figure 5a shows how the power dissipation in individual films varies with increasing wavelength. As we transition from visible to NIR wavelengths, dissipation decreases in the TiN and increases in the AZO. This redistribution demonstrates increasing light-matter interaction with AZO the faster of the two materials.

Plotting the temporal response of the device at different probe wavelengths shows a gradual decrease in the overall relaxation time with increasing wavelength (Fig. 5b). The effective temporal

response of the device can be modeled as a weighted sum of the individual responses of the materials:

$$\left(\frac{\Delta R}{R}\right)(t) = \boldsymbol{\alpha}\left(A\mathrm{e}^{\frac{-t}{\tau 1}} + B\mathrm{e}^{\frac{-t}{\tau 2}}\right) + \boldsymbol{\beta}D\mathrm{e}^{\frac{-t}{\tau}} + \boldsymbol{\Gamma} \qquad (1)$$

where $\boldsymbol{\alpha}$ is the weighted contribution of TiN (with the 2-time constants $\tau 1$ & $\tau 2$) and $\boldsymbol{\beta}$ is the weighted contribution of the AZO (single time constant $\tau$). $\boldsymbol{\Gamma}$ is an offset attributed to slower thermal effects and noise due to probe fluctuations. Supplementary Table S4 has the fitting parameters. The ratio of $\boldsymbol{\alpha}/(\boldsymbol{\alpha} + \boldsymbol{\beta})$, the ratio of the absorption in TiN to the total absorption, decreases as expected with increasing wavelength (Supplementary Fig. S3). Intuitively, as the probe wavelength increases, more light interacts with the AZO versus TiN. Thus, AZO dictates the modulation dynamics, resulting in faster

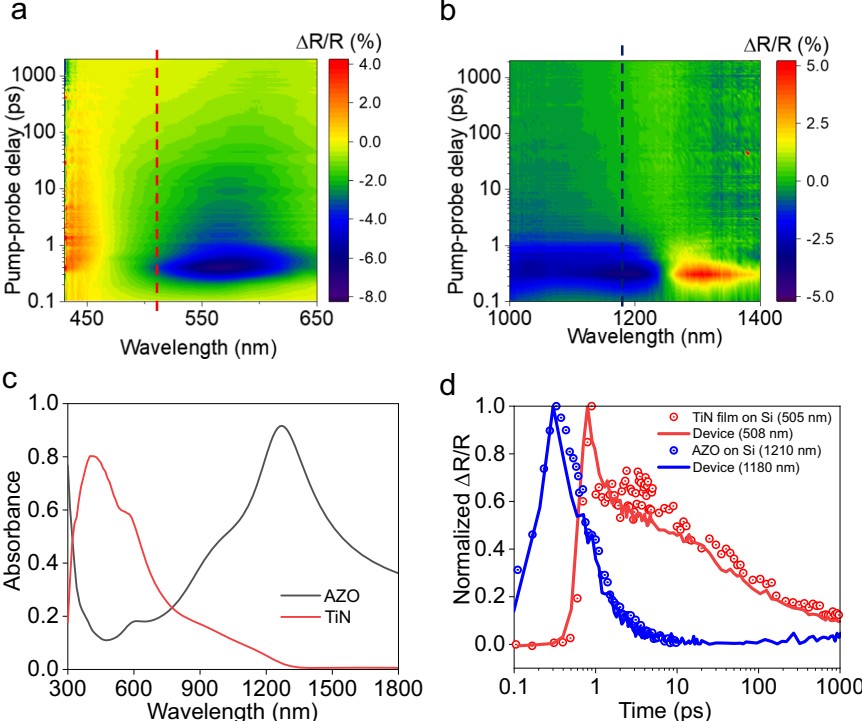

**Fig. 4 | Switching response of the device. a** Transient reflectance modulation versus time for the absorber with visible probes and (**b**) near-infrared probes. The red and blue vertical dashed lines correspond to the device dynamics at 508 nm and 1180 nm, respectively. These are plotted in Fig. 4d as solid lines. **c** The absorbance of the p-polarized light in the device at an angle of 50o as calculated by the Finite Element Method, showing that light interacts strongly with individual films near their respective ENZ points. **d** The transient modulation response of the device at a probe wavelength of 508 nm (solid red line) and 1180 nm (solid blue line). The points are the modulation responses of the TiN film at 505 nm (red points) and the AZO film at 1210 nm (blue points). The slower TiN dynamics are shifted by 0.4 ps for clarity.

response dynamics. Therefore, it is possible to control the switching speed of the same device from nanosecond to picosecond scale by simply changing the wavelength of operation A more accurate model of the relaxation response can be developed by incorporating the wavelength-dependent relaxation dynamics of the individual materials, the magnitude of reflectance modulation in the individual materials, the resonance shifts, ultrafast non-equilibrium dynamics of hot-electrons in the materials and interfaces, the effect of surface roughness on carrier recombination, and other nonlinearities triggered by the strong pump.

Supplementary Information Section S7 shows our experimental studies of other modes of switching speed control employing the findings of this study. In addition to the wavelength control, for instances where the probe wavelength is fixed, the speed can be changed by adjusting other parameters, such as the angle of incidence (Supplementary Fig. S4) and the polarization of probe beam, or the pump-wavelength (Supplementary Fig. S5) which changes the light-matter interaction with the constituent materials.

## Discussion

While extensive strides have been conducted in the dynamic control of the phase, amplitude, and polarization of light with optical devices, there exists a gap in engineering the temporal dynamics of tunable systems. In this work, by employing an all-optical double-layer switch comprising both fast and slow-responding materials, we demonstrated advanced control over the temporal dynamics of such a hybrid all-optical switch. The device supports two p-polarized light resonances – a radiative visible ENZ mode in the TiN layer and an infrared Ferrell-Berreman mode in the AZO layer. When excited by the same 325 nm wavelength pump, close to these resonances, the device has a switching speed under 10 ps when probed at 1180 nm and over 2 ns

when probed at 508 nm. The probe follows the relaxation dynamics of the constituent material near its epsilon near zero point, where the light-matter interaction is the strongest. By sweeping the probe between these ENZ-related resonances, one can operate the same device at switching speeds spanning two orders of magnitude, enabling a great degree of control over the switching speed.

The proposed approach utilizes slower materials that are very robust and allow to enhance the field intensities while faster materials ensure an ultrafast dynamic response. We show that fast permittivity modulation can be obtained even when materials with relatively slow dynamics are used as a back reflector, to enhance the modulation at the desired operational wavelength. Our findings underline the importance of conducting broadband pump-probe characterization of the full device stack, including spacers, back reflectors, and substrates in addition to the active layer when designing all-optical switches. Furthermore, when designing complex time-varying devices utilizing multiple materials, a good understanding of the dynamic response of each constituent material is required to achieve the desired performance.

We show that the overall temporal response of a dual-material all-optical switch can be represented by a weighted sum of the relaxation behaviors of each of the constituent materials. The zero-to-zero response of an all-optical switch at a particular wavelength is dictated by the material with the stronger light-matter interaction, in other words, by the material that confines the most incident light within it. By changing the probe-light wavelength to interact with different materials in the device, the range of operating speeds and thus bandwidth can be varied by two orders of magnitude. We experimentally demonstrated that controlling the probe-light interaction with the individual TiN or AZO layers enables the large variation of the observed speed. Moreover, we mapped the light-matter interaction of the probe

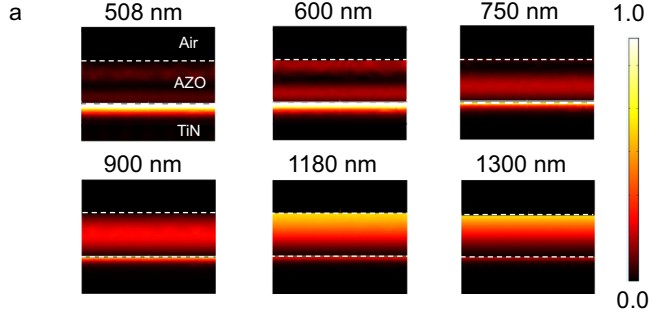

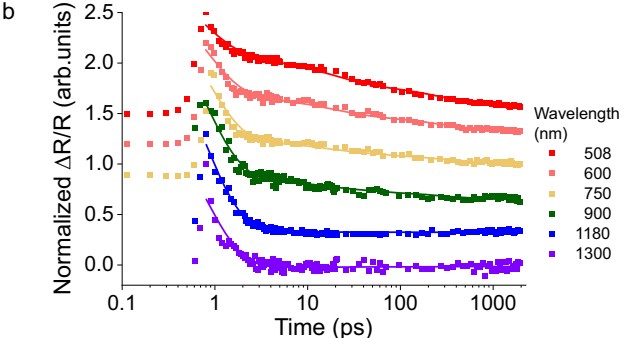

**Fig. 5 | Light-matter interaction dictates the switching response. a** Simulated power dissipation (normalized) in the films computed using measured optical properties of the films. The white dashed lines denote the Air-AZO and AZO-TiN boundaries. As the wavelength increases, more light interacts with the AZO film compared to TiN. **b** Experimentally obtained reflectance modulation versus pump-probe delay time. The modulation for each wavelength is normalized to the maximum response at each wavelength and shifted to the same time for easier comparison. Starting from 1300 nm (purple), the datasets for each wavelength are vertically shifted by 0.3 from the next (from purple to red) for easier comparison. The points represent experimental results, and the solid lines represent fitted values. As the probe wavelength increases, the relaxation rate transitions gradually from nanosecond to picosecond-scales.

with the materials and correlate their relative absorbances with particular relaxation times. Finally, we demonstrated that further control over the switching speed can be attained by controlling the probe polarization, angle of incidence, as well as the pump wavelength.

Our approach can be used to mix and match materials with significantly different temporal dynamics to achieve multiple switching speeds and bandwidths within the same platform for multiband data transmission. In the field of all-optical switching beyond Moore's Law, optical switches have been demonstrated to operate at femtosecond speeds, offering terahertz or higher switching rates[48]. However, in most practical applications, such switches would still need to communicate with electrical components, whose speed is limited to gigahertz scales. To mitigate the modulation strength versus bandwidth compromise, it is desirable to have the response speed of the modulating effect be only a couple of times faster than the operating speed of the device one is trying to get[67]. Thus, the option of tunable switching speeds that match slowly increasing electronic switching speeds offers a way to bridge the gap between electronic and optical communications.

Robust, laser-tolerant materials can be also employed as back reflectors in optical switching schemes and nonlinear optical experiments. Their slower speeds do not affect the ultrafast response of the active medium. Examples of such systems include nonlinear optical phenomena such as frequency shifts[68], negative refraction[64], time refraction[29], and photonic time crystals[69] that require ultrafast changes

in the material permittivity at fast, femtosecond timescales, while operating at large laser powers.

Finally, our method can be used to extract the temporal dynamics of individual materials in complex, multilayer structures by controlling where the probe light is strongly confined.

Harnessing control over the all-optical switching speeds by developing a holistic approach to studying the temporal dynamics of time-varying devices and incorporating the contribution of each material will be crucial in developing time-varying devices for telecommunications optics and other nonlinear optical applications.

## Data availability
The source data that supports this study is available from Figshare (https://doi.org/10.6084/m9.figshare.23734116).

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

## Acknowledgements

The authors acknowledge support by the U.S. Department of Energy, Office of Basic Energy Sciences, Division of Materials Sciences and

Engineering under Award DE-SC0017717 (TCO materials growth and characterization), the Office of Naval Research under Award N00014-20-1-2199 (TCO optical characterization), and the Air Force Office of Scientific Research under Award FA9550-20-1-0124 (transition metal nitrides studies). Use of the Center for Nanoscale Materials, an Office of Science user facility, was supported by the U.S. Department of Energy, Office of Science, Office of Basic Energy Sciences, under Contract No. DE-AC02-06CH11357.

## Author contributions

S.S. conceived the basic idea for this work. S.S., M.G.O., and Z.K. performed the COMSOL simulations. Z.K. performed the TMM calculations. S.S. and S.N.C. fabricated and characterized the device. B.D. designed the preliminary experimental set-up for the pump-probe spectroscopy. S.S., B.D., and R.D.S. carried out the measurements. S.S., A.V.K., A.B., and S.P. analyzed the experimental results. R.D.S., Z.J., A.V.K., V.M.S., and A.B. supervised the research and the development of the manuscript. S. S. and M.G.O. wrote the first draft of the manuscript. All co-authors subsequently took part in the revision process and approved the final copy of the manuscript.

## Competing interests

The authors declare no competing interest.
