## [Peer Review File · Nature Communications]

REVIEWER COMMENTS

Reviewer #1 (Remarks to the Author):

In the work: 'Engineering the Temporal Dynamics of All-Optical Switching with 'Fast' and 'Slow' Materials', S. Saha and colleagues perform pump-probe spectroscopy measurements on a bi-layer system placed on a silicon (Si) substrate composed of titanium nitride (TiN) and aluminum doped zinc oxide (AZO) of 130 nm and 250 nm thickness, respectively. In the main experiment, the sample is pumped at a wavelength of 325 nm (pulse fluence 1.5 mJ/cm²) and probed in the visible (450 nm – 650 nm) and near-infrared (800 nm – 1600 nm). The authors report a time-dependent relative reflectivity modulation with a zero-to-zero decay time that depends on the probe wavelength, ranging from the ~ps (@1.21 μ m) to the ~ns (@ 505 nm). The claim is that such a platform allows the all-optical tuning of the response time from THz to GHz.

The main idea is that the interband pump pulse excites carriers that modify the electric permittivity of both TiN and AZO. Scattering processes eventually restore the permittivity's static value but do so on different timescales, ~ns for TiN and ~ps for AZO. When such materials are probed close to the ENZ point (where the real part of the permittivity crosses zero), they display the strongest interaction with a transient behavior associated with their respective timescale. In this work, the two materials are stacked such that the overall response can be tuned from slow (~ns) to fast (~ps) by tuning the wavelength from the TiN to the AZO ENZ point. The chosen configuration, TiN at the bottom and AZO on top, allows a strong interaction with TiN at shorter wavelengths (where AZO shows low losses) and AZO at longer wavelengths (where TiN is metallic and reflects the incident probe light).

The manuscript is well-written, technically sound, and the explanation is clear. However, I have concerns about its suitability for publication in Nature Communications. My main reasons are the following:

1. The manuscript gives the impression of aiming at engineering the all-optical switching response time. However, the reported results show the slow vs. fast time response only for two stacked flat films. Moreover, such films behave very similarly when probed individually (Fig. 3) or when stacked (Fig. 1). Therefore, it is not clear where control/optimization/engineering comes into play.
2. The possibility of modulating the response time is interesting, but it can only be achieved at different wavelengths. In this sense, it is unclear how straightforward it would be to apply this concept to applications where one aims to modulate a control signal at a fixed wavelength. In other words, one can adjust the response time, but only by changing one property of the signal (the wavelength) one would arguably like to maintain unchanged in the first place.
3. It is mentioned that a model has been developed, but the reported analysis only shows an expression with the fitted exponential functions that include the timescales of the two materials. Several works on all-optical switching are detailed with accurate models (see more in the next point).
4. The manuscript does not mention or discuss several contributions to the field of all-optical switching. For example, *Advanced Materials* 32, 2020 (<https://doi.org/10.1002/adma.201806317>) gives a good summary of nonlinear optical metasurfaces for all-optical switching. Ultrafast all-optical switching has

been, in fact, investigated quite intensively in the last five years or so. Silicon metasurfaces have been accurately modeled and experimentally demonstrated to achieve ps modulation, see ACS Photonics, 4, 2017 (<http://dx.doi.org/10.1021/acsp Photonics.7b00544>). GaAs metasurfaces have also been shown to achieve even faster modulation, see Nature Communications, 8, 2017 (DOI: 10.1038/s41467-017-00019-3). Besides the intrinsic timescale of the relaxation processes, the spatial reconfiguration of hot carriers has also been exploited for sub-ps all-optical switching, see Nature Photonics, 14, 2020 (<https://doi.org/10.1038/s41566-020-00702-w>) and Advanced Optical Materials, 10, 2022 (DOI: 10.1002/adom.202102549). The use of an ENZ region has also been proposed in ITO, see Optics Express, 30, 2022 (<https://doi.org/10.1364/OE.457875>).

For these reasons, at this stage, I cannot recommend the publication in Nature Communications.

Additional comments

5. Page 4: it is mentioned that AZO has \sim fs response if pumped intraband. What about TiN? Can modulation frequency be extended more by changing the pump frequency?

6. What are typical values for the non-normalized transient reflectances reported in this work, e.g. for Fig. 1d (especially to see if the modulation changes for different probe wavelengths) and Fig. 3c, e? Knowing how much the baseline reflectivity can be modulated would be useful.

Reviewer #2 (Remarks to the Author):

This work of Saha et al deals with the temporal dynamics of all-optical switching by probing a bilayer of a slow optical material, i.e. a conductive material with low mobility of electrons manifested in the temporal variation of reflectance, such as TiN with a fast optical material (i.e. of high electron mobility) such as AZO, in wavelengths between the corresponding ENZs of the constituent layers.

The work is a continuation of the previous report of the same authors in Advanced Materials (DOI: 10.1002/adma.202109546), however, there is no lack of novelty as it is not just an incremental work, but it provides really new physics by studying the different temporal responses of the bilayer switch by varying the wavelength and provides new engineering aspects and perspectives for applications. Therefore, it is suitable for Nature Comms.

The work is well written, carefully implemented, and the results are sound and well documented.

Only one point seems to lack some clarity. In particular, in Fig. 1c the redshift of the pumped TiN is not explained adequately. Since the density of conduction electrons is temporarily increasing during pumping there would be redshift of both the TiN and AZO deeps. It is implied that the pumped electrons in TiN are more localized but there is no suggestion of a clear mechanism for that (e.g. the density of conduction electrons in TiN is already too high so the additional pumped electrons would reduce their mobility, in contrast to the AZO case). Given that it is quite well known that TiN and AZO are conductors of low and high electron mobility, respectively, it might be useful to attempt to correlate the fast and slow optical character with the electron mobility.

RESPONSE TO REVIEWER COMMENTS

Reviewer #1 (Remarks to the Author):

In the work: 'Engineering the Temporal Dynamics of All-Optical Switching with 'Fast' and 'Slow' Materials', S. Saha and colleagues perform pump-probe spectroscopy measurements on a bi-layer system placed on a silicon (Si) substrate composed of titanium nitride (TiN) and aluminum doped zinc oxide (AZO) of 130 nm and 250 nm thickness, respectively. In the main experiment, the sample is pumped at a wavelength of 325 nm (pulse fluence 1.5 mJ/cm²) and probed in the visible (450 nm – 650 nm) and near-infrared (800 nm – 1600 nm). The authors report a time-dependent relative reflectivity modulation with a zero-to-zero decay time that depends on the probe wavelength, ranging from the ~ps (@1.21 μ m) to the ~ns (@ 505 nm). The claim is that such a platform allows the all-optical tuning of the response time from THz to GHz.

The main idea is that the interband pump pulse excites carriers that modify the electric permittivity of both TiN and AZO. Scattering processes eventually restore the permittivity's static value but do so on different timescales, ~ns for TiN and ~ps for AZO. When such materials are probed close to the ENZ point (where the real part of the permittivity crosses zero), they display the strongest interaction with a transient behavior associated with their respective timescale. In this work, the two materials are stacked such that the overall response can be tuned from slow (~ns) to fast (~ps) by tuning the wavelength from the TiN to the AZO ENZ point. The chosen configuration, TiN at the bottom and AZO on top, allows a strong interaction with TiN at shorter wavelengths (where AZO shows low losses) and AZO at longer wavelengths (where TiN is metallic and reflects the incident probe light).

Author response: We thank the Reviewer for the constructive criticism of our manuscript. We would like to clarify that the main focus of our work is to demonstrate how strong light-matter interaction with a specific material at specific wavelengths can be used to control the operating speed of a simple multi-material switch. We have performed more experiments to strengthen our claim and address the concerns. Changes in the revised version of the manuscript are highlighted in yellow.

The manuscript is well-written, technically sound, and the explanation is clear. However, I have concerns about its suitability for publication in Nature Communications. My main reasons are the following:

1. The manuscript gives the impression of aiming at engineering the all-optical switching response time. However, the reported results show the slow vs. fast time response only for two stacked flat films. Moreover, such films behave very similarly when probed individually (Fig. 3) or when stacked (Fig. 1). Therefore, it is not clear where control/optimization/engineering comes into play.

Author response:

This work aims to demonstrate that within the same system (or stack), controlled light interaction with different materials results in different switching speeds (at certain wavelengths). We designed TiN and AZO films such that they behave similarly in terms of dynamics when probed separately (Fig. 3) or when probed as a stacked structure (Fig. 1). This allows for control of the

device bandwidth by engineering light-matter interaction in a particular constituent material. In our experiment, the control comes from light interacting with different materials, so the constituent materials needed the same response when probed individually (i.e., TiN on Si, and AZO on Si, shown in Section 5) or in a stacked configuration (TiN-AZO switch, shown in Section 6).

While we illustrate this approach through the wavelength selection, similar control can be achieved by changing the probe polarization, the angle of incidence, and/or the pump wavelength. Controlling each of these parameters could change the light-matter interaction with different materials. We illustrate this further in response to the next question.

To demonstrate our concept, the constituent materials (AZO and TiN) were specifically chosen for their distinct epsilon-near-zero (ENZ) wavelengths and the sharp contrast in their relaxation speeds^{1,2}. Therefore, we first studied the two materials separately with the same pump-probe conditions in a single device, demonstrating how the individual responses merge in the switch, resulting in different operational speeds.

The layered stack was chosen to prevent fabrication-induced imperfections that can alter the recombination time, as shown in previous work³.

The engineered device (stack) response comes from a combination of the responses of the individual materials controlled by the probe light interaction. The addition of the TiN and AZO material responses result in the combined device response, as shown in Section 7.

2. The possibility of modulating the response time is interesting, but it can only be achieved at different wavelengths. In this sense, it is unclear how straightforward it would be to apply this concept to applications where one aims to modulate a control signal at a fixed wavelength. In other words, one can adjust the response time, but only by changing one property of the signal (the wavelength) one would arguably like to maintain unchanged in the first place.

Author response: We thank the Reviewer for this question; it encouraged us to perform more experiments to improve the clarity of the paper's message. In this work, the strong wavelength dependence of the device response time is used to illustrate how the light-matter interaction affects the device speed. For instances where the probe wavelength is fixed, the speed can be changed by adjusting several parameters, such as the incident light polarization, pump wavelength, or the angle of incidence. We have conducted further experiments to confirm this and have included the results in the supporting information section. In the visible wavelength range, where most of the light will interact with the TiN layer, resulting in a slow (ns) response, a fast (ps) response can still be achieved by utilizing a different pump that interacts more strongly with and modulates the faster material, resulting in a picosecond response. We have included the results in response to question 5 and the supporting information.

Manuscript, Section 7:

In addition to the wavelength control, for instances where the probe wavelength is fixed, the speed can be changed by adjusting other parameters, such as the angle of

incidence, the polarization of the probe beam or the pump wavelength, which changes the light-matter interaction with the constituent materials. We have conducted further experiments to confirm this and have included the results in the supporting information section S7.

Supporting information S7

In our system, for a fixed wavelength, changing the **probe angle of incidence** decreases the proportion of light interacting with the fast (AZO) material and increases the interaction with the slow (TiN) material, thus slowing the effective switching rate down (Fig. S4a).

Similarly, we also observe a nanosecond tail in the transient response with an **s-polarized probe** at the NIR wavelength range (Fig. S4b).

Figure S4. (a) At an angle of 30 degrees, a greater proportion of the probe interacts with the TiN, resulting in a nanosecond tail (b) A similar, slower response can also be seen by moving from a p-polarized to an s-polarized probe at the same wavelength. There is also a sign reversal of the signals at a femtosecond timescale that can be attributed to complex, non-equilibrium dynamics of hot electrons, which are beyond the scope of this study and require further investigation.

- It is mentioned that a model has been developed, but the reported analysis only shows an expression with the fitted exponential functions that include the timescales of the two materials. Several works on all-optical switching are detailed with accurate models (see more in the next point).

Author response: This investigation aimed to highlight how the dynamics of two materials can be combined (engineered) to get a device temporal response (with signatures from both materials), where the device response is a combination of the response of individual materials. Hence, our simple model serves to develop an intuitive picture of how the dynamics of the two different materials add up to form the dynamic response of the stack, as observed by the probe at various wavelengths.

Detailed models of the electron dynamics of the materials under the strong pumps have been derived in prior studies, both for TiN^{2,4} and AZO^{5,6}, and are beyond the scope of this investigation. Our work highlighted the probe interaction with different materials. We have clarified section 7 of the manuscript with the study's limitations.

“A more accurate model of the relaxation response can be developed by incorporating the wavelength-dependent relaxation dynamics of the individual materials, the magnitude of the reflectance modulation in the individual materials, the resonance shifts, the ultrafast, non-equilibrium dynamics of hot-electrons in the materials and the interfaces, the effect of surface roughness on carrier recombination, and other nonlinearities triggered by the strong pump.”

4. The manuscript does not mention or discuss several contributions to the field of all-optical switching. For example, *Advanced Materials* 32, 2020 (<https://doi.org/10.1002/adma.201806317>) gives a good summary of nonlinear optical metasurfaces for all-optical switching. Ultrafast all-optical switching has been, in fact, investigated quite intensively in the last five years or so. Silicon metasurfaces have been accurately modeled and experimentally demonstrated to achieve ps modulation, see *ACS Photonics*, 4, 2017 (<http://dx.doi.org/10.1021/acsp Photonics.7b00544>). GaAs metasurfaces have also been shown to achieve even faster modulation, see *Nature Communications*, 8, 2017 (DOI: 10.1038/s41467-017-00019-3). Besides the intrinsic timescale of the relaxation processes, the spatial reconfiguration of hot carriers has also been exploited for sub-ps all-optical switching, see *Nature Photonics*, 14, 2020 (<https://doi.org/10.1038/s41566-020-00702-w>) and *Advanced Optical Materials*, 10, 2022 (DOI: 10.1002/adom.202102549). The use of an ENZ region has also been proposed in ITO, see *Optics Express*, 30, 2022 (<https://doi.org/10.1364/OE.457875>).

For these reasons, at this stage, I cannot recommend the publication in *Nature Communications*.

Author response: We agree that the ultrafast optical switching field has seen a recent increase in interest. The omissions of some of the key contributions were unintentional, and we apologize for them. We thank the Reviewer for highlighting the omitted publications; we have added them to the manuscript's relevant sections.

“There have been numerous studies on modulating the permittivity of materials in various ways, including electrical^{35,40-43}, thermal⁴⁴⁻⁴⁷, and optical methods^{9,10,14,48-57}. These studies have focused on the scaling of modulation with power^{26,42,58,59}, enhancement with engineered structures^{8,14}, and modulation limits^{45,60,61}.

Thus far, few studies have focused on engineering the switching speed of tunable and tailorable devices⁶², which is generally fixed and/or defined by the response times of the material constituents.”

But we again emphasize that in the review paper (*Adv. Mat.* 32, 2020), most switching speed alterations happen during device fabrication. The novelty of our method lies in changing the

operational speed post-fabrication by changing parameters that can be controlled without physically altering the device.

Additional comments

5. Page 4: it is mentioned that AZO has ~fs response if pumped intraband. What about TiN? Can modulation frequency be extended more by changing the pump frequency?

Author response: With AZO, the modulation speed can be controlled by switching from an intraband (100s of fs) to an interband (few ps) pump. The relaxation time of titanium nitride has been investigated with various pump wavelengths^{2,4}. A fast fs scale response is observed at high powers, but the zero-to-zero response time is dominated by the slow phonon-phonon relaxation, spanning nanoseconds². As a result, the response time of TiN films cannot be made faster by using a different pump response.

However, the overall speed of the switch can indeed be made faster by utilizing a pump that is mostly absorbed by the AZO in the stack. The following experiment shows the device response a probe with a visible wavelength of 488 nm (50° angle of incidence) when pumped with an intraband pump with $\lambda = 1400$ nm (p-polarized, 70° angle of incidence). The AZO absorbs most of the signal at this configuration, resulting in a fast, femtosecond response. We have added these results to Supporting Information S7.

“At the visible wavelengths, where most of the light will primarily interact with the TiN layer, a fast response can still be achieved by utilizing a different pump which interacts more strongly with the faster (AZO) material, thus leading to a sub-picosecond response. The following experiment shows the device response with a visible probe at 488 nm wavelength (at 50° angle of incidence) when pumped with an intraband pump of 1400 nm wavelength (p-polarized, 70° angle). At this configuration, the AZO absorbs most of the signal, resulting in a fast, femtosecond response (Fig. S5, solid black line). Note that the TiN response time at the same probe wavelength is in nanoseconds, regardless of the pump wavelength (dashed lines).

Figure S5. An intraband pump at 1400 nm and a 70° angle of incidence results in a femtosecond device response to a visible probe at 488 nm. This is because the pump is mostly absorbed in the AZO layer, resulting in a femtosecond device response, even though most of the probe interacts with the TiN layer. The dashed lines show the responses of the TiN film (red) with a 1400 nm pump, showing the slower dynamics and the device dynamics with a 325 nm pump. “

6. What are typical values for the non-normalized transient reflectances reported in this work, e.g. for Fig. 1d (especially to see if the modulation changes for different probe wavelengths) and Fig. 3c, e? Knowing how much the baseline reflectivity can be modulated would be useful.

Author response: At the pump fluence of 1.5 mJ/cm² used in this work, the individual TiN layer modulates by a $\Delta R/R$ of 5% and AZO by 15% (Fig 3). For the stack, the modulation peaks at 8% near the visible and 5 % near the infrared wavelength range (Fig. 4).

The modulation depth of the materials depends on various factors, such as the pump wavelength and power. In other works, modulations of 15% have been demonstrated with TiN films², and up to 40% with AZO films^{1,5}. Patterning the materials can further increase the modulation depth and reduce the power requirements⁷, but was avoided in this work as patterning would alter the recombination rates in the materials by adding additional recombination sites³. We chose the thicknesses specifically to ensure that both TiN and AZO layers absorb approximately equal proportions of the 325 nm pump.

Reviewer #2 (Remarks to the Author):

This work of Saha et al deals with the temporal dynamics of all-optical switching by probing a bilayer of a slow optical material, i.e. a conductive material with low mobility of electrons manifested in the temporal variation of reflectance, such as TiN with a fast optical material (i.e. of high electron mobility) such as AZO, in wavelengths between the corresponding ENZs of the constituent layers.

The work is a continuation of the previous report of the same authors in *Advanced Materials* (DOI: 10.1002/adma.202109546), however, there is no lack of novelty as it is not just an incremental work, but it provides really new physics by studying the different temporal responses of the bilayer switch by varying the wavelength and provides new engineering aspects and perspectives for applications. Therefore, it is suitable for *Nature Comms*.

The work is well written, carefully implemented, and the results are sound and well documented.

Only one point seems to lack some clarity. In particular, in Fig. 1c the redshift of the pumped TiN is not explained adequately. Since the density of conduction electrons is temporarily increasing during pumping there would be redshift of both the TiN and AZO dips. It is implied that the pumped electrons in TiN are more localized but there is no suggestion of a clear mechanism for that (e.g. the density of conduction electrons in TiN is already too high so the additional pumped electrons would reduce their mobility, in contrast to the AZO case). Given that it is quite well known that TiN and AZO are conductors of low and high electron mobility, respectively, it might be useful to attempt to correlate the fast and slow optical character with the electron mobility.

Author response: We thank the Reviewer for the positive assessment of our manuscript. Regarding the red shift in titanium nitride, an interband pump raises the density of free electrons and the Drude damping factor due to lattice heating. However, at a pump fluence of 1.5 mJ/cm^2 , about 10^{20} cm^{-3} e-h pairs are generated in TiN. The intrinsic carrier density of TiN is about 10^{22} cm^{-3} . Since the density of free electrons increases only by a small fraction compared to the initial density, the plasma frequency remains almost the same, whereas, due to lattice heating, the increase in the Drude damping factor plays the dominant role, raising the overall permittivity of TiN. This effect is the most viable explanation for the redshift in TiN. Similar effects are also seen in Au and ZrN under a pump of 3.1 eV^2 . AZO has an intrinsic carrier concentration of 10^{20} cm^{-3} ; as a result, under the same pump fluence, the carrier density almost doubles, resulting in a large change in the plasma frequency compared to the Drude damping factor, resulting in a negative change in the permittivity, and a blue shift. At this stage, we are unable to comment on the relationship between the interband relaxation dynamics in the different materials and their electron mobility.

References:

1. Kinsey, N. *et al.* Epsilon-near-zero Al-doped ZnO for ultrafast switching at telecom wavelengths. *Optica* **2**, 616–622 (2015).
2. Diroll, B. T., Saha, S., Shalaev, V. M., Boltasseva, A. & Schaller, R. D. Broadband Ultrafast Dynamics of Refractory Metals: TiN and ZrN. *Adv Opt Mater* **8**, 2000652 (2020).
3. Shcherbakov, M. R. *et al.* Ultrafast all-optical tuning of direct-gap semiconductor metasurfaces. *Nature Communications* 2017 8:1 **8**, 1–6 (2017).
4. George, H. *et al.* Nonlinearities and carrier dynamics in refractory plasmonic TiN thin films. *Opt Mater Express* **9**, 3911 (2019).
5. Clerici, M. *et al.* Controlling hybrid nonlinearities in transparent conducting oxides via two-colour excitation. *Nat Commun* **8**, 15829 (2017).
6. Bohn, J. *et al.* All-optical switching of an epsilon-near-zero plasmon resonance in indium tin oxide. *Nat Commun* **12**, 1017 (2021).
7. Ren, M., Cai, W. & Xu, J. Tailorable Dynamics in Nonlinear Optical Metasurfaces. *Advanced Materials* **32**, 1806317 (2020).

REVIEWER COMMENTS

Reviewer #1 (Remarks to the Author):

I reviewed the reply to the comments and re-read the manuscript. I am satisfied with the additional explanations, references and experiments they have performed. In particular, I found interesting the change in response time by tuning parameters other than the wavelength. There is only one thing I suggest the author to discuss about the new Fig. S4. In panel a, it looks like the 30 deg cases flatten fast for both wavelengths but somehow the differential reflectivity remains non-zero for a long time (even more than the scale allows to see). Similarly in panel b the s-polarizations show a $\Delta R/R$ going to zero after about 1 ps before increasing for longer times. Can the author comment on the slow time dynamics of the red curves and the total time before the differential signals go back to zero? Provided the authors include such discussion, I can recommend the publication.

REVIEWER COMMENTS

Reviewer #1 (Remarks to the Author):

I reviewed the reply to the comments and re-read the manuscript. I am satisfied with the additional explanations, references and experiments they have performed. In particular, I found interesting the change in response time by tuning parameters other than the wavelength.

Author response: We thank the Reviewer for the positive comment. The further experiments encouraged by the review helped make our work stronger and more relevant. We have made the following changes in the manuscript to reflect the newer experiments.

“Abstract

...This work demonstrates that the zero-to-zero response time of an all-optical switch can instead be varied through controlled light-matter interaction in so-called “fast” and “slow” materials in a single device. Probed in the epsilon-near-zero (ENZ) operational regime of plasmonic titanium nitride, a material with a slow response time, the switch exhibits a relatively slow, nanosecond response time. The response time decreases as the probe wavelength increases, reaching the picosecond time scale when the hybrid device is probed in the ENZ-regime of the aluminum-doped zinc oxide, the faster material. Overall, the response time of the switch is shown to vary by two orders of magnitude in a single device. It can be selectively controlled through the interaction of light with the constituent materials. The ability to adjust the switching speed by controlling the light-matter interactions in a multi-material structure provides several additional degrees of freedom in the design of all-optical switches, whereby the speed of a switch can be adjusted post-fabrication by controlling the probe’s polarization, angle of incidence, wavelength, and also the pump-wavelength.”

Discussion and conclusion:

Moreover, we mapped the light-matter interaction of the probe with the materials and correlate their relative absorbances with particular relaxation times. Finally, we demonstrated that further control over the switching speed can be attained by controlling the probe polarization, angle of incidence, as well as the pump wavelength.

There is only one thing I suggest the author to discuss about the new Fig. S4. In panel a, it looks like the 30 deg cases flatten fast for both wavelengths but somehow the differential reflectivity remains non-zero for a long time (even more than the scale allows to see).

Author response: In Fig. S4a, for the 30° angle of incidence, more of the electromagnetic field interacts with TiN, the dynamics thus follow the slower TiN dynamics, which take several nanoseconds to eventually fall to zero. As specified in the paper, the device relaxation is a weighted sum of the relaxation rates of the two materials, and has signatures of both decays. For the 30° case, the non-zero term takes several nanoseconds to fall to zero because of the slow TiN response^{1,2}.

We have added the explanation in the text.

“In Fig. S4a, for the 30° angle of incidence, more of the electromagnetic field interacts with TiN, the dynamics thus follow the slower TiN dynamics, which take several nanoseconds to eventually fall to zero. For the 30° case, the non-zero term takes several nanoseconds to fall to zero because of the slow TiN response^{6,7}.”

Similarly in panel b the s-polarizations show a $\Delta R/R$ going to zero after about 1 ps before increasing for longer times. Can the author comment on the slow time dynamics of the red curves and the total time before the differential signals go back to zero? Provided the authors include such discussion, I can recommend the publication.

Author response: The dynamics in Fig. S4b are more complicated. AZO relaxes fast, decreasing the overall reflectance change, coming to a minimum at 1 ps, after which the response follows the relaxation rate in TiN, which spans several nanoseconds. The sign reversal and the increase after 1 ps result from the shift and the broadening of additional Fabry-Perot resonances due to the injection of photoexcited hot-carriers, which interact with the reflectance signal. The complex interplay of resonance shift and broadening that result in the tens of picosecond to hundred picosecond component likely arises from hot carrier injection as well as thermal expansions that alter resonant conditions. In this case, s-polarization presents higher sensitivity to these changes. We have earlier published a tutorial on the transient reflectance and transmittance spectra at

epsilon near zero films, with comprehensive explanations for such sign changes and anomalous decay behavior in the pump-probe reflectance studies of ENZ films³. Additionally, we have added the explanation, and the reference in the supporting information.

“...The sign reversal and the increase after 1 ps result from the shift and the broadening of additional Fabry-Perot resonances due to the injection of photoexcited hot-carriers, which interact with the reflectance signal. The complex interplay of resonance shift and broadening that result in the tens of picosecond to hundred picosecond component likely arises from hot-carrier injection as well as thermal expansions that alter resonant conditions. In this case, s-polarization presents higher sensitivity to these changes. A more rigorous investigation of similar dynamics in ENZ films is reported in the work by Fruhling et al⁸.”

Figure S4. (a) At an angle of 30 degrees, a greater proportion of the probe interacts with the TiN, resulting in a nanosecond tail (b) A similar, slower response can also be seen by moving from a p-polarized to an s-polarized probe at the same wavelength. There is also a sign reversal of the signal change that can be attributed to complex, non-equilibrium dynamics of photoinjected carriers that result in the shift and broadening of additional resonances in the structure.

References:

1. Diroll, B. T., Saha, S., Shalaev, V. M., Boltasseva, A. & Schaller, R. D. Broadband Ultrafast Dynamics of Refractory Metals: TiN and ZrN. *Adv Opt Mater* **8**, 2000652 (2020).
2. George, H. *et al.* Nonlinearities and carrier dynamics in refractory plasmonic TiN thin films. *Opt Mater Express* **9**, 3911 (2019).
3. Fruhling, C., Ozlu, M. G., Saha, S., Boltasseva, A. & Shalaev, V. M. Understanding all-optical switching at the epsilon-near-zero point: a tutorial review. *Appl Phys B* **128**, 1–12 (2022).